# SLMRec: Distilling Large Language Models into Small for Sequential Recommendation

**Wujiang Xu**[1], **Qitian Wu**[2], **Zujie Liang**[3], **Jiaojiao Han**[4],
**Xuying Ning**[5], **Yunxiao Shi**[6], **Wenfang Lin**[3], **Yongfeng Zhang**[1*]

[1] Rutgers University    [2] Eric and Wendy Schmidt Center, Broad Institute of MIT and Harvard
[3] Ant Group    [4] Dian Diagnostics Group Co.
[5] University of Illinois Urbana-Champaign    [6] University of Technology Sydney

## Abstract

Sequential Recommendation (SR) task involves predicting the next item a user is likely to interact with, given their past interactions. The SR models examine the sequence of a user's actions to discern more complex behavioral patterns and temporal dynamics. Recent research demonstrates the great impact of LLMs on sequential recommendation systems, either viewing sequential recommendation as language modeling or serving as the backbone for user representation. Although these methods deliver outstanding performance, there is scant evidence of the necessity of a large language model and how large the language model is needed, especially in the sequential recommendation scene. Meanwhile, due to the huge size of LLMs, it is inefficient and impractical to apply a LLM-based model in real-world platforms that often need to process billions of traffic logs daily. In this paper, we explore the influence of LLMs' depth by conducting extensive experiments on large-scale industry datasets. Surprisingly, our motivational experiments reveal that most intermediate layers of LLMs are redundant, indicating that pruning the remaining layers can still maintain strong performance. Motivated by this insight, we empower small language models for SR, namely SLMRec, which adopt a simple yet effective knowledge distillation method. Moreover, SLMRec is orthogonal to other post-training efficiency techniques, such as quantization and pruning, so that they can be leveraged in combination. Comprehensive experimental results illustrate that the proposed SLMRec model attains the best performance using only 13% of the parameters found in LLM-based recommendation models, while simultaneously achieving up to 6.6x and 8.0x speedups in training and inference time costs, respectively. Besides, we provide a theoretical justification for why small language models can perform comparably to large language models in SR. The source code and datasets are available at the URL [1].

## 1 Introduction

Learning temporal interest information is fundamental for sequential recommendation models. Traditional sequential recommendation (**TSR**) methods (Wu et al., 2017; Hidasi et al., 2015; Kang & McAuley, 2018; Sun et al., 2019) focus on the development of intricate sequential encoders, evolving from LSTM and GRU architectures to the self-attention layers and Transformer models. However, the state-of-the-art performance in TSR has hit a plateau, limited by model sizes that usually feature fewer than 0.1 billion parameters.

Recently, Large Language Models (LLMs) (Achiam et al., 2023; Touvron et al., 2023; Anil et al., 2023) have made significant advancements in various aspects by scaling the size of the training data or the model's architecture. Building upon the scaling laws delineated in prior research (Kaplan et al., 2020; Hoffmann et al., 2022), it endows LLMs with enhanced expressivity, culminating in superior performance benchmarks. Naturally, a burgeoning trend among contemporary LLM-based recommendation architectures has raised concerns. The current LLM-based recommender system

---

[*] Corresponding author.

[1] https://github.com/WujiangXu/SLMRec

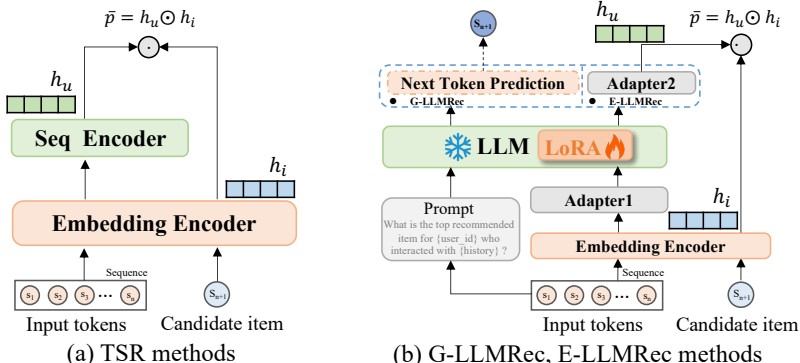

Figure 1: This overview compares traditional sequential recommendation (TSR) methods with LLM-based recommendation (LLMRec) methods. Here, $h_u$ and $h_i$ represent the user and item representations, respectively. In contrast to G-LLMRec methods, E-LLMRec approaches adhere to the TSR prediction framework. These methods leverage LLMs as feature extractors in the manner of BERT, diverging from the generative focus of G-LLMRec.

can be classified as 1) **generation-based approaches**, *e.g.*, P5 (Geng et al., 2022; Xu et al., 2023a), CoLLM (Zhang et al., 2023b) and LLaRa (Liao et al., 2023); 2) **embedding-based approaches** such as E4SRec (Li et al., 2023a), CLLM4Rec (Zhu et al., 2023) and Lite-LLM4Rec (Wang et al., 2024). As shown in Fig. 1, generation-based approaches (**G-LLMRec**) encode an item as a token and formulate the sequential recommendation as the next token prediction task. By contrast, embedding-based approaches (**E-LLMRec**) regard the last hidden representation as user representation and learn an external adapter to compute user-item preference. The adoption of LLMs has vastly driven the development of sequence recommendation tasks, bringing an improvement of nearly 20% against the TSR model on the benchmark (Li et al., 2023a; Liao et al., 2023; Wang et al., 2024). This arouses the following research motivation for this work.

• Some researchers (Ardalani et al., 2022; Zhang et al., 2023a; 2024) have attempted to investigate the scaling laws in the recommendation domain. However, the largest model examined in these studies is less than 1 billion parameters, significantly smaller than the 175 billion parameters of GPT-3 (Brown et al., 2020). Additionally, the focus has been primarily on test loss rather than on ranking-based evaluation metrics, which limits the practical applicability of their findings. Recent studies (Liang et al., 2023; Gromov et al., 2024; Men et al., 2024) on the NLP domain suggest a high degree of redundancy in the LLMs' model architecture. Since the ID information of the recommendation domain has not been explicitly learned during the LLMs' training process, we also aim to find out whether increasing the model size of LLMs is beneficial for the SR task.

• Despite the large performance gain, the LLMRec methods also escalate the model size significantly, *e.g.*, nearly 70 times greater parameters compared with TSR models (from 0.1B to 7B+). Even within the parameter-efficient training technique (Hu et al., 2021a), the paradigm still poses a significant challenge for real-world sequential recommendation use cases, where billions of traffic logs every day and potential new items need to be processed constantly. This disparity imposes strict hardware demands and makes it both inefficient and infeasible to deploy the LLMRec model.

**Our contributions.** This paper presents an initial attempt to reassess the need for LLMs in sequential recommendation. To explore the reasons for the significant improvement of LLMRec methods, we conduct a series of experiments on large-scale industry datasets to investigate the effects of reducing the number of parameters during the training and inference stages on overall performance. From the empirical results, we found some profound insights that the improvement of the rise of the model parameters is not consistent. Meanwhile, it reveals that some layers of LLMs are redundant in the recommendation task, similar to findings in NLP domains (Men et al., 2024; Gromov et al., 2024).

Motivated by these findings, we empower small language models for the sequential recommendation, named SLMREC. We adopt the vanilla knowledge distillation approaches to align the representation knowledge. Moreover, multiple supervision signals are crafted to steer the student model toward acquiring task-aware knowledge within its hidden representations. Additionally, our model operates without the need for any supplementary model design elements and is compatible with other quantization and pruning techniques utilized within LLMs.

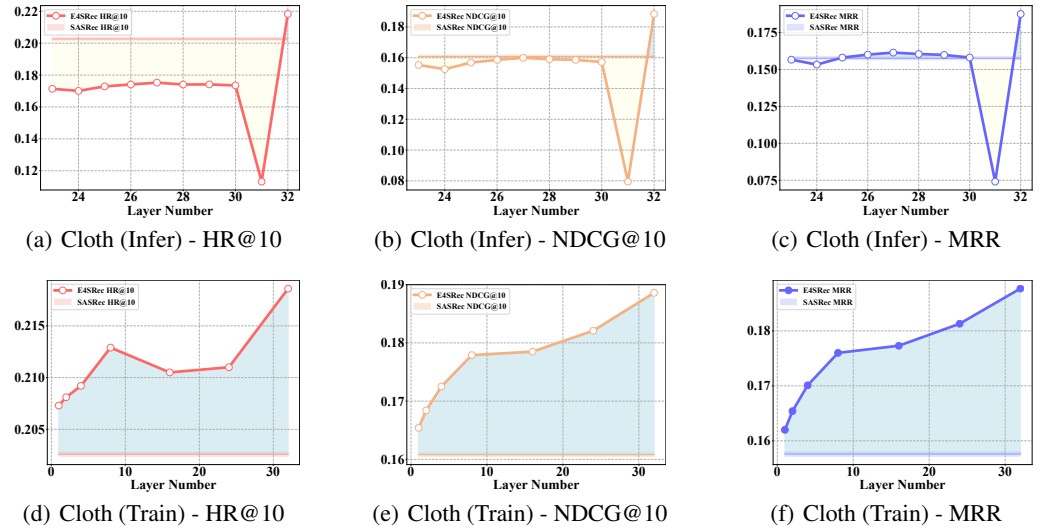

Figure 2: We present the relationship between the number of decoder layers and the final recommendation performance, with the performance of SASRec plotted as a baseline. Figures (a)-(c) show the results of directly using representations from the middle layers for inference without training, while (d)-(f) prune the later layers and train a model using only the specified number of layers. From the results, we observe that deeper decoder layers introduce redundancy in recommendation tasks, with models utilizing fewer layers (8-layer) achieving performance nearly equivalent to (24-layer) models.

Extensive experiments have revealed that SLMRec, with a model size of less than 1 billion parameters, can deliver performance that is remarkably competitive with baselines using LLMs sized over 7 billion parameters. Furthermore, SLMRec achieves up to 6.6x/8.0x speedup in terms of training/inference time costs against LLM-based recommendation models. Besides, we present the results of SLMRec employing online knowledge distillation, demonstrating its competitive performance. Beyond emprical experiment results, we provide a theoretical justification for why small language models can perform comparably to large language models in SR.

## 2   MOTIVATIONAL EXPERIMENTS

As described above, here we try to explore the effectiveness of LLMs in recommendation via decreasing the parameters of popular LLMs (*i.e.*, LLaMa-7B) and observe the change in performance.

**Evaluation Protocol.** In the motivational experiment, we select SASRec as a traditional sequential recommendation baseline due to its performance (Klenitskiy & Vasilev, 2023). We adopt[2] embedding-based method (Li et al., 2023a) as the baseline, named E4SRec, to easily generate the ranking for the full/sampled list of items. As shown in Fig. 2, a pre-trained embedding layer learned from SASRec is used to obtain the sequential item embedding. Then we concatenate the item embeddings with the prompt embeddings obtained after the tokenization. After encoding of stacked attention blocks of LLM, we regard the representation of the last layers as the user representation. Then, we follow the TSR methods to calculate the inner product of user embeddings and item embeddings from the pre-trained embedding layer to serve as the score for the user-item pair. Also, cross-entropy loss and fully candidate item are utilized for the optimization to achieve best results (Xu et al., 2024a; Petrov & Macdonald, 2023). To reduce both computational demands and processing time, LoRA (Hu et al., 2021a) is used to update a comparatively smaller set of parameters. Besides, to generate an unbiased evaluation for fair comparison (Krichene & Rendle, 2020; Zhao et al., 2020), we randomly sampled 999 negative items, which were items not interacted with by the user, along with 1 positive item that served as the ground-truth interaction. To obtain large-scale industry data, we use the Amazon 18 version[3] dataset in this paper. More details are shown in Section 5.

---

[2]To accelerate and align with the prediction head of traditional SR methods, we remove the original softmax layer and instead use the dot product of the user and item representations to compute the prediction score.

[3]https://nijianmo.github.io/amazon/index.html

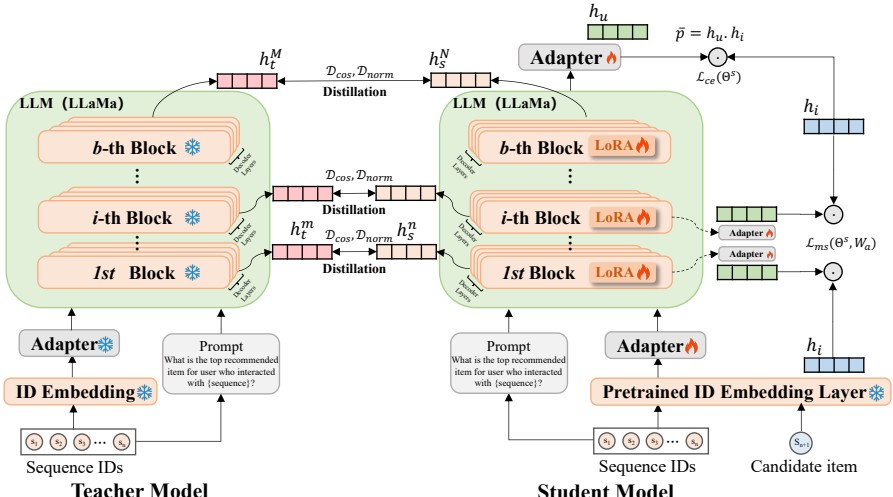

Figure 3: The overview of SLMREC. A layer-wise knowledge distillation approach is applied to align the representation knowledge by grouping the layer into serveral blocks. The teacher and student model share a similar E-LLMRec model architecture. Multiple supervision signals are introduced to steer the student model toward acquiring fine-grained task-aware knowledge.

**Evaluation Strategy.** To examine the connection between the number of parameters and the performance of LLM-based methods (E4SRec), we have truncated the original LLM architecture—in this case, a 32-layer decoder from the LLaMa 7B model—by pruning the decoder layers during both the inference and the training stages. As a direct inference method, we refrain from additional training using new labels and instead directly employ the output from the final ten layers as user representations to gauge recommendation performance. Instead of direct inference, we focus on conserving the initial layers of the decoder and proceed to train a more lightweight E4SRec model while adhering to the original training protocol. The models resulting from varying levels of layer retention are designated as E4SRec$_l$, with the variable $l$ indicating the number of layers retained. The chosen values of $l$ encompass a spectrum, specifically $\{1, 2, 4, 8, 16, 24, 32\}$. Results from both experimental approaches are graphically depicted in Figure 2, providing insight into how the models' depth influences their recommendation capabilities.

**Insights.** From Figure 2 (a)-(b), we can observe that directly utilizing the representation of other layers without training cannot obtain a comparative performance. Compared to TSR baseline SASRec, Figure 2 (c)-(d) yield the following insightful findings: (1) As the number of layers increases, the performance of the model also improves. Furthermore, even when the model has the same layer number (*i.e.*, $l$=2) as SASRec, its performance is still superior to that of SASRec. We assume the gains observed in LLM-based methods could likely be attributed to the larger hidden representation size ((*i.e.*, 4096 V.S. 128), the initialization from LLMs, and the introduction of PEFT (Hu et al., 2021a). (2) When $l$ is set ranging from 8-24, the model's improvement is slight. It reveals that an 8-layer E4SRec$_8$ can obtain nearly as informative user representations as a 24-layer E4SRec$_{24}$. Considering the two findings above, it naturally inspires us to explore better training methods to obtain a smaller-size LLM-based SR model that is comparable with large models. If we want to learn a E4SRec$_M$ that perform similar as E4SRec$_N$ (M < N), we should make sure the intermediate representations in E4SRec$_M$ to be as closer to those in E4SRec$_N$ as possible. Knowledge distillation (KD) is a straightforward idea in this case. Thus, we design a simple yet effective knowledge distillation method to train a tiny LLM-based model with similar performance. For the motivation experiment results in Movie domain, it can be found in Appendix B.1. In Section 6, we provide a theoretical justification that aligns with these empirical insights.

## 3 PRELIMINARIES

In this study, rather than constructing complex additional structures, we slightly modify existing E-LLMRec methods for our purposes. Initially, we delineate the E-LLMRec model that we employ for sequential recommendation tasks.

**Model structure.** The E-LLMRec models capitalize on an ID embedding layer from TSR models such as BERT4Rec, SASRec, and GRU4Rec, which is pre-trained on a designated dataset (Sun et al., 2019; Kang & McAuley, 2018; Hidasi et al., 2015). The objective of sequential recommendation is to forecast subsequent items utilizing the user action sequence $S = (i_1, i_2, ..., i_T)$, a sequence that is either truncated or padded to maintain uniform length. Through truncation and padding, we derive the user's action sequence mask, serving as the attention mask in LLMs (Large Language Models). The fixed-length sequence $S \in \mathbb{R}^T$ is translated into a sequential representation $\mathbf{S} \in \mathbb{R}^{T \times d_0}$ via the pre-trained ID embedding layer. A linear transformation is then applied to upscale the representation from a lower dimension $d_0$ to a higher dimension $d_1$ suitable for the hidden layers within the LLMs.

Upon defining the prompt template, the tokenization layer within the LLMs processes the natural language input into corresponding text embeddings and their associated attention masks. These embeddings and attention masks, derived from both the ID sequence and the text, are then introduced into the LLM decoder. The final temporal output $\mathbf{h}_M$ from the last layer of the decoder is inferred as the user representation and subsequently mapped through a linear layer to condense the dimensionality from $d_1$ back to $d_0$. Finally, user-item interaction predictions $\bar{p}$ are inferred by executing a dot product between the user and item representations. The learning process of the model is refined through the application of a cross-entropy loss.

$$\hat{p}_i = \frac{e^{\bar{p}_i}}{\sum_{j \in I} e^{\bar{p}_j}}; \quad \mathcal{L}_{ce}(\Theta_s) = -\sum_{u \in U, i \in I} y_{(ui)} log(\hat{p}_i). \tag{1}$$

where $U$ and $I$ denote the whole user set and item set. $y_{ui}$ denotes user-item interaction label.

**Knowledge Distillation.** Knowledge distillation is a technique aimed at transferring knowledge from a sophisticated teacher model to a more streamlined student model (Hinton et al., 2015). We represent the teacher by $f_t(\Theta_t)$ and the student by $f_s(\Theta_s)$. We aim to solve the following optimization problem:

$$\min_{\Theta_s} [\mathcal{L}_{ce}(\Theta_s) + \mathcal{D}_{kd}(\Theta_t, \Theta_s)]. \tag{2}$$

Here, $\mathcal{D}_{kd}(\Theta_t, \Theta_s)$ signifies the knowledge distillation loss, which quantifies the discrepancies between the teacher and the student models. A prevalent method involves employing the KL divergence to evaluate the divergence between the logits produced by both models. One well-established training schema is known as offline distillation, wherein the teacher is fully trained beforehand and remains unchanged, while the student is refined based on the criteria outlined in Eq. 6. In the offline knowledge distillation manner, the teacher model $\Theta_t$ is initially trained in a designated training set by minimizing the cross-entropy loss $\mathcal{L}_{ce}$.

# 4 SLMRec

In this work, we do not adopt logits-based knowledge distillation, as our goal is for the student model to learn how to encode hidden representations similar to the teacher model, rather than merely replicating its predictions. To achieve this, we perform feature distillation across multiple layers. Specifically, considering that the teacher model consists of $M$ stacked decoder layers and the student model has $N$ stacked decoder layers, we design several feature regularizers to guide the distillation process at regular intervals between the hidden representations of both models. We divide the layers of the teacher and student models into blocks by grouping every $m$ layers of the teacher and every $n$ layers of the student. The number of resulting blocks is $B$, calculated as $B = \lfloor \frac{M}{m} \rfloor = \lfloor \frac{N}{n} \rfloor$. Let the hidden representations from the teacher model be denoted as: $\mathbf{H}_t = \{\mathbf{h}_t^m, \ldots, \mathbf{h}_t^M\}$, where $\mathbf{h}_t^m$ represents the final temporal dimension of the hidden representation from the $m$-th layer of the teacher. Similarly, the hidden representations from the student model are denoted as: $\mathbf{H}_s = \{\mathbf{h}_s^n, \ldots, \mathbf{h}_s^N\}$. In this study, we use a deeper LLM as the teacher model and a shallower LLM as the student model, both sharing the same hidden dimension $d$, such that $\mathbf{H}_t, \mathbf{H}_s \in \mathbb{R}^{B \times d}$.

**Feature Similarity.** To regulate the alignment of feature directions between the teacher and student models, we employ a cosine similarity-based loss term. Formally, it is described by the equation:

$$\mathcal{D}_{\cos}(\Theta_t, \Theta_s) = \frac{1}{B} \sum_{k=1}^{B} \frac{\mathbf{h}_t^{(km)} \cdot \mathbf{h}_s^{(kn)}}{\|\mathbf{h}_t^{(km)}\|_2 \cdot \|\mathbf{h}_s^{(kn)}\|_2}. \tag{3}$$

**Feature Norm Regularization.** In addition, we introduce a straightforward regularization term designed to minimize the L2 distance between the hidden representations of the teacher and student models. It is mathematically formulated as:

$$\mathcal{D}_{norm}(\Theta_t, \Theta_s) = \frac{1}{B} \sum_{k=1}^{B} \|\mathbf{h}_t^{(km)} - \mathbf{h}_s^{(kn)}\|_2^2. \tag{4}$$

**Multiple Supervision.** Furthermore, we employ multiple supervision strategies to steer the student model toward assimilating specific aspects of recommendation-related knowledge. For each representation, we learn additional adapters ( $W_a$ ) to reduce the dimension. The modified prediction ( $\hat{p}_t^{(km)}$ ) can be acquired as described by Eq. 1:

$$\mathcal{L}_{ms}(\Theta_s, W_a) = \frac{1}{B-1} \sum_{k=1}^{B-1} \mathcal{L}_{ce}(y, \hat{p}_t^{(km)}). \tag{5}$$

**Total Loss.** Integrating the aforementioned distillation losses, the composite objective function for training the student model is given by:

$$\min_{\Theta_s, W_a} [\mathcal{L}_{ce}(\Theta_s) + \lambda_1(1 - \mathcal{D}_{cos}(\Theta_t, \Theta_s)) + \lambda_2 \mathcal{D}_{norm}(\Theta_t, \Theta_s) + \lambda_3 \mathcal{L}_{ms}(\Theta_s, W_a)]. \tag{6}$$

where $\lambda_1$, $\lambda_2$ and $\lambda_3$ are hyperparameters that control the contribution of each term.

## 5 EXPERIMENTS

In this section, we present extensive experiments to demonstrate the effectiveness of SLMREC, aiming to answer the following four research questions (**RQs**).

• **RQ1**: How does the performance of our proposed SLMREC model compare to LLM-based recommendation models when evaluated on a large-scale industry dataset?

• **RQ2**: What is the comparative efficiency and runtime of our SLMREC model against the G-LLMRec and E-LLMRec models?

• **RQ3**: Whether the proposed three knowledge regularizers work?

• **RQ4**: Is it feasible to train our model, SLMREC, simultaneously with an untrained teacher model?

### 5.1 EXPERIMENT SETUP

For our experimental evaluation, we utilize data from the clothing, movies, music, and sports categories within the extensive, industry-scale Amazon18 dataset[4]. Statistics of the datasets are shown in Table 1. In all datasets, we interpret any rating above 3 as positive feedback, indicating user interaction with the item, and employ timestamps to establish the chronological order of actions. We eliminate users and items that have fewer than 5 associated actions to ensure sufficient data density. The historical sequence

Table 1: Statistics of the Amazon datasets. $|\mathcal{U}|$, $|\mathcal{V}|$, and $|\mathcal{E}|$ denote the number of users, items, and ratings, respectively.

| Dataset | $|\mathcal{U}|$ | $|\mathcal{V}|$ | $|\mathcal{E}|$ | Density |
|---|---|---|---|---|
| Cloth | 1,219,678 | 376,858 | 11,285,464 | 0.002% |
| Movie | 297,529 | 60,175 | 3,410,019 | 0.019% |
| Music | 112,395 | 73,713 | 1,443,755 | 0.017% |
| Sport | 332,447 | 12,314 | 146,639 | 0.008% |

of interactions for each user is divided into three segments: (1) the most recent interaction is reserved for testing, (2) the second most recent for validation, and (3) all preceding interactions are used for training. Based on the ranking results, we utilize the typical *top-N* metrics hit rate (HR@{1, 5, 10}), normalized discounted cumulative gain (NDCG@{5,10}) (Järvelin & Kekäläinen, 2002)

[4]https://nijianmo.github.io/amazon/index.html

Table 2: Experimental results (%) on the Cloth and Movie dataset. The missing MRR value of Open-P5 is unavailable due to the time complexity constrictions. The number on the left of the arrow is the layers $N$ of the student model. The left number on the right of the arrow is the layers $M$ of the teacher model. For Open-P5, we adopt LLaMa as their backbone. We highlight the methods with the **best** and second-best average performances. We also give the average ranking of each evaluation metric. Moreover, $E4SRec_4$, which has the same number of layers as our SLMREC, is also marked.

| Model | Cloth | | | | Movie | | | | Rank |
|---|---|---|---|---|---|---|---|---|---|
| | **HR@1** | **HR@5** | **NDCG@5** | **MRR** | **HR@1** | **HR@5** | **NDCG@5** | **MRR** | **Rank** |
| Caser | 9.66 | 15.18 | 12.66 | 13.03 | 4.27 | 14.96 | 9.57 | 10.36 | 13.50 |
| GRU4Rec | 13.79 | 15.46 | 14.64 | 15.15 | 10.56 | 19.47 | 15.11 | 15.46 | 9.25 |
| BERT4Rec | 13.60 | 14.66 | 14.14 | 14.59 | 9.68 | 14.91 | 12.40 | 12.74 | 11.63 |
| SASRec | 13.08 | 16.94 | 15.01 | 15.76 | 5.57 | 16.80 | 11.17 | 12.08 | 11.63 |
| HGN | 15.96 | 18.70 | 17.30 | 18.27 | 7.54 | 19.20 | 13.42 | 14.73 | 6.50 |
| LightSANs | 14.12 | 20.32 | 17.30 | 16.86 | 6.08 | 17.54 | 11.81 | 12.82 | 8.00 |
| $S^3$-Rec | 14.10 | 18.67 | 16.10 | 16.95 | 7.75 | 20.39 | 15.69 | 14.34 | 7.50 |
| DuoRec | 13.06 | 18.29 | 15.79 | 15.42 | 10.07 | 20.37 | 17.96 | 16.61 | 7.88 |
| MAERec | 13.29 | 18.35 | 15.68 | 16.13 | 8.89 | 20.24 | 16.03 | 15.28 | 8.38 |
| Open-P5 | 14.13 | 17.68 | 17.02 | - | 12.66 | 21.98 | 17.13 | - | 5.67 |
| E4SRec | 16.71 | 19.45 | 18.09 | 18.77 | 14.74 | 23.79 | 19.45 | 19.74 | 1.75 |
| $E4SRec_8$ | 15.30 | 18.54 | 16.91 | 17.60 | 13.32 | 22.49 | 17.99 | 18.46 | 4.00 |
| $E4SRec_4$ | 14.58 | 18.05 | 16.32 | 17.01 | 11.80 | 21.54 | 16.73 | 17.20 | 5.75 |
| $SLMREC_{4\leftarrow8}$ | **16.69** | **19.47** | **18.07** | **18.74** | **15.29** | **24.25** | **19.90** | **20.36** | **1.50** |

and Mean Reciprocal Rank (MRR) (Sarwar et al., 2001) to evaluate the model performance. For all the metrics, higher values indicate better performance. Models that achieve the highest MRR performance on the validation set, including ours and other baseline models, will be preserved for subsequent performance evaluation on the test set. In order to ensure an unbiased evaluation, we adopt the methodology employed in previous works (Krichene & Rendle, 2020; Zhao et al., 2020), wherein we randomly select 999 negative items (i.e., items that the user has not interacted with) and combine them with 1 positive item (i.e., a ground-truth interaction) to form our recommendation candidates for the ranking test. Detailed hyperparameters of our model in each dataset are in Appendix B.2.

## 5.2 PERFORMANCE COMPARISONS

**Compared Methods.** We compare our method with three classes of baselines: (1) Traditional sequential recommendation methods, i.e., GRU4Rec (Hidasi et al., 2015), Caser (Tang & Wang, 2018), HGN (Ma et al., 2019), BERT4Rec (Sun et al., 2019), SASRec (Kang & McAuley, 2018) and LightSANs (Fan et al., 2021). (2) Self-supervised sequential recommendation methods, i.e., $S^3$-Rec (Zhou et al., 2020), DuoRec (Qiu et al., 2022) and MAERec (Ye et al., 2023). (3) G-LLMRec method: Open-P5$_{LLaMa}$[5] (Xu et al., 2023a). (4) E-LLMRec method: E4SRec (Li et al., 2023a). A detailed introduction to these baselines can be found in Appendix B.3. It should be noted that we did not select various G-LLMRec methods or E-LLMRec methods as baselines. This is because the differences between each LLM-based method are minimal, and our model is a universal approach that is not confined to a specific model type. Our primary focus is to improve the efficiency of language model utilization. Hence, we opted to select one G-LLMRec method (Open-P5) and one E-LLMRec method (E4SRec) as baselines.

**Quantitative Results (RQ1).** Tables 2–3 showcase the quantitative comparison of four large-scale sequential recommendation datasets. From our analysis, we have several insightful observations: (1) LLM-based recommendation methods exhibit substantial improvements over traditional sequential recommendation (TSR) methods, primarily due to their enhanced modeling capacity which adeptly extracts informative sequential interest patterns. (2) Our model, $SLMRec_{4\leftarrow8}$, outperforms the teacher model $E4SRec_8$ by leveraging knowledge distillation within the hidden layers. By refraining from applying this constraint prior to the prediction phase, we enable the final representation to organically gravitate towards the label—yielding an approximate 8% enhancement in performance in comparison to the teacher model. (3) Introducing vanilla knowledge distillation techniques into LLMRec, without altering the model structure, allows $SLMRec_{4\leftarrow8}$ to achieve a marginally superior performance compared to $E4SRec_{32}$. This suggests that small language models equipped with

---

[5]For Open-P5, we adopt the version of LLaMa as the foundation model in their code repository implementation to ensure the best results are achieved.

Table 3: Experimental results (%) on the Music and Sport dataset.

| Model | Music | | | | Sport | | | | Rank |
|---|---|---|---|---|---|---|---|---|---|
| | HR@1 | HR@5 | NDCG@5 | MRR | HR@1 | HR@5 | NDCG@5 | MRR | |
| Caser | 0.71 | 3.28 | 1.96 | 2.29 | 1.05 | 3.75 | 2.39 | 2.84 | 13.50 |
| GRU4Rec | 1.89 | 3.22 | 2.57 | 3.08 | 5.26 | 7.75 | 6.52 | 7.08 | 10.13 |
| BERT4Rec | 2.10 | 3.16 | 2.64 | 3.11 | 4.81 | 6.70 | 5.79 | 6.26 | 10.63 |
| SASRec | 1.82 | 5.72 | 3.79 | 4.51 | 4.70 | 8.43 | 6.59 | 7.24 | 8.75 |
| HGN | 2.01 | 5.49 | 3.82 | 4.17 | 3.42 | 6.24 | 4.83 | 5.30 | 10.50 |
| LightSANs | 1.05 | 4.06 | 2.54 | 3.00 | 5.18 | 8.94 | 7.07 | 7.72 | 8.25 |
| $S^3$-Rec | 2.48 | 7.37 | 4.94 | 4.68 | 4.14 | 8.49 | 6.89 | 7.35 | 6.88 |
| DuoRec | 1.84 | 4.50 | 3.19 | 3.04 | 4.13 | 8.81 | 7.03 | 6.64 | 9.13 |
| MAERec | 2.19 | 6.35 | 4.67 | 3.96 | 4.01 | 8.35 | 6.65 | 6.98 | 8.63 |
| Open-P5 | 4.35 | 8.12 | 6.74 | - | 5.49 | 8.50 | 6.92 | - | 5.33 |
| E4SRec | **5.62** | **9.29** | **7.50** | **7.98** | **6.40** | **9.67** | **8.05** | **8.70** | **1.75** |
| E4SRec$_8$ | 5.46 | 8.86 | 7.21 | 7.74 | 5.48 | 8.63 | 7.06 | 7.76 | 3.63 |
| E4SRec$_4$ | 5.33 | 8.75 | 7.08 | 7.59 | 5.41 | 8.65 | 7.04 | 7.72 | 4.50 |
| SLMRec$_{4\leftarrow8}$ | **5.72** | **9.15** | **7.48** | **8.03** | **6.62** | **9.83** | **8.25** | **8.89** | **1.25** |

Table 4: Experiment results (%) of ablation study.

| SLMREC | Cloth | | | | Movie | | | |
|---|---|---|---|---|---|---|---|---|
| | HR@1 | HR@5 | NDCG@5 | MRR | HR@1 | HR@5 | NDCG@5 | MRR |
| $+\mathcal{D}_{cos}$ | 16.10 | 18.85 | 17.48 | 18.17 | 14.83 | 23.08 | 19.08 | 19.45 |
| $+\mathcal{D}_{cos},\mathcal{D}_{norm}$ | 16.28 | 19.12 | 17.69 | 18.40 | 14.86 | 23.89 | 19.36 | 19.84 |
| $+\mathcal{D}_{cos},\mathcal{L}_{ms}$ | **16.85** | 19.05 | 17.96 | 18.59 | 15.05 | 23.48 | 19.40 | 19.76 |
| $+\mathcal{D}_{cos},\mathcal{D}_{norm},\mathcal{L}_{ms}$ | 16.69 | **19.47** | **18.07** | **18.74** | **15.29** | **24.25** | **19.90** | **20.36** |

Table 5: Efficiency comparison of Open-P5, E4SRec, and our SLMREC in terms of epoch-wise training time (hours), inference time (hours), number of training parameters (B) and inference parameters (B). These comparisons were conducted on a machine with an A100 GPU. The training batch size for all models was standardized at 256. During inference, E4SRec and SLMREC utilized a batch size of 512, whereas Open-P5's inference was performed with a batch size of 1.

| Method | Tr time(h) | Inf time(h) | Tr params (B) | Inf params (B) |
|---|---|---|---|---|
| Open-P5$_{LLaMa}$ | 0.92 | 4942 | 0.023 | 7.237 |
| E4SRec | 3.95 | 0.415 | 0.023 | 6.631 |
| **SLMREC$_{4\leftarrow8}$** | 0.60 | 0.052 | 0.003 | 0.944 |

efficacious training strategies can rival, or even exceed, larger language models in the sequential recommendation task. This phenomonon is also matched with our therotical justification in Section 6.

**Model Efficiency (RQ2).** We report the time efficiency and parameters of comparative baselines and our model in Table 5. All time and parameter metrics represent the average across the four datasets reported. Inference time evaluates the prediction ranking among 1,000 candidate items for each user. Detailed training and inference times for each dataset are provided in Appendix B.4. The Open-P5, an LLMRec model based on generative methods, offers a reasonable training duration. Yet, during the inference phase, it becomes considerably time-consuming (4942 hours) as it necessitates generating a substantial pool of candidate items (for instance, 1000). Owing to the intrinsic workings of generative LLMs, employing generation-based LLMRec models for the comprehensive ranking of extensive item sets is not advised. Our model outperforms E4SRec with enhanced efficiency, maintaining only 13% and 14% in E4SRec's parameters for training and inference, respectively. Moreover, our SLMREC demonstrates a remarkable gain in speed, being 6.6 times faster during training and 8.0 times quicker in inference than E4SRec.

**Ablation Study (RQ3).** As shown in Table 4, SLMREC, when enhanced with various knowledge regularizers (namely $\mathcal{D}_{cos}$, $\mathcal{D}_{norm}$ and $\mathcal{L}_{ms}$), demonstrates improved performance. The regularizers $\mathcal{D}_{cos}$ and $\mathcal{D}_{norm}$ aid SLMREC in aligning its intermediate representations with those of the teacher model, thereby endowing it with more potent representational extraction capabilities. Meanwhile, $\mathcal{L}_{ms}$ steers the model to assimilate domain knowledge pertinent to recommendation systems within its preliminary layers. The ablation study in Music and Sport domain can be found in Appendix B.5.

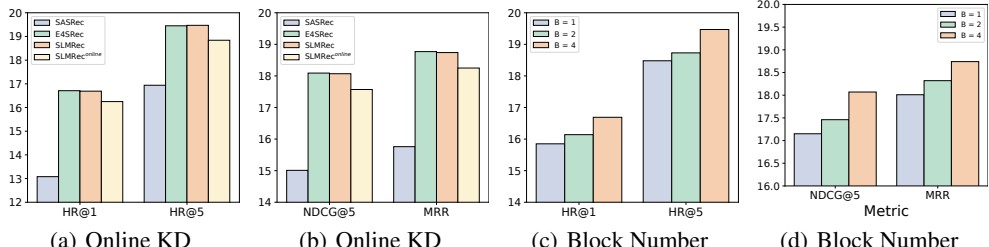

| (a) Online KD | (b) Online KD | (c) Block Number | (d) Block Number |

Figure 4: Experiment results (%) of online KD and block number $B$ in the Cloth dataset.

## 5.3 MODEL STUDY

**Study of Online KD (RQ4).** In our methodology, we first train the teacher model on downstream recommendation tasks and then train the student model through knowledge distillation, which is an offline knowledge distillation technology. In this section, we demonstrate that we can train both the teacher model and SLMREC together on downstream recommendation tasks, which constitutes an online knowledge distillation. Under this setting, we are able to achieve comparative results.

**Study of block number $B$.** We also conducted experiments to investigate the effect of block number $B$. As shown in Figure 4, when $B$ is set to 4, our model achieves the best performance. When $B$ is set to 1 or 2, the feature constraint imitation for each block within SLMREC is diminished relative to the teacher model, resulting in a decline in performance.

## 6 THEORETICAL JUSTIFICATIONS

Beyond empirical experiments, we aim to provide insights into why small language models can perform as effectively as large language models in learning desirable user representations. Specifically, we focus on the feature propagation process within a single layer of an LLM, as outlined below:

$$\mathbf{H}^{(k)} = \mathbf{H}^{(k-1)} + \mathbf{A}^{(k)}\mathbf{H}^{(k-1)}, \tag{7}$$

where $\mathbf{H}^{(k)}$ represents the hidden representation of the $k$-th layer, and $\mathbf{A}^{(k)}$ is the attention matrix. In LLaMa, the attention matrix is defined as $\mathbf{A} = \mathrm{softmax}\left(\frac{\mathbf{Q}'\mathbf{K}'^{\top}}{\sqrt{d_k}}\right)$, where $\mathbf{Q}'$ and $\mathbf{K}'$ incorporate rotational encoding (Su et al., 2024). Our analysis is hinged on interpreting the stack of propagation layers of Transformers as optimization dynamics for minimizing energies of certain forms (Shuman et al., 2013; Kalofolias, 2016; Fu et al., 2022; Wu et al., 2024).

**Proposition 1.** *Given the updating matrix $\hat{\mathbf{A}}^{(k)} = \mathbf{A}^{(k)} + \mathbf{I}$, Eqn. 7 is equivalent to a gradient descent step with respect to the following optimization problem:*

$$\min_{H} \left\| \mathbf{H} - \hat{\mathbf{A}}^{(k)}\mathbf{H}^{(k-1)} \right\|_2^2 \tag{8}$$

As $\mathbf{A}^{(k)}$ changes across layers, multi-layer attention models can be interpreted as a series of iterative descent steps, each focusing on layer-specific denoising objectives. We will show that this multi-layer structure can be simplified into a single-layer model while retaining the same denoising effectiveness.

**Proposition 2.** *For any $K$-layer attention model (where $K$ is an arbitrary positive integer) with the layer-wise updating rule defined by Eqn. 7, there exists $\mathbf{C}^*$ such that one gradient descent step for the optimization problem (from the initial embeddings $\mathbf{H}^{(0)}$)*

$$\min_{\mathbf{H}} \left\| \mathbf{H} - \mathbf{C}^*\mathbf{H}^{(0)} \right\|_2^2, \tag{9}$$

*where $\mathbf{C}^*$ associated with $\mathbf{A}$, can yield the output embeddings $\mathbf{H}^{(K)}$ of the $K$-layer model.*

These findings indicate that for any multi-layer stacked decoder, an equivalent single-layer decoder can be constructed to encode hidden representations in a similar way. Moreover, while the multi-layer model optimizes distinct objectives at each layer, this may introduce redundancy when compared to a single-layer model that achieves its objective in a single step. Consistent with the motivation underlying our framework design, we employ knowledge distillation (KD) to guide the one-layer network, enabling it to streamline the learning process and replicate the feature extraction capabilities of a multi-layer network.

## 7  RELATED WORK

In this section, we introduce the most related background and scientific investigations to this work, which are roughly divided into five categories, *i.e.*, 1) Sequential Recommendation, 2) Knowledge Distillation (KD), 3) Depth-wise Knowledge of LLMs, 4) Model Pruning, and 5) Parameter-Efficient Fine-Tuning (PEFT). For details on sections three through five, please refer to Appendix C.

**Sequential Recommendation.** Traditional Sequential Recommendation (TSR) methods (Wu et al., 2017; Hidasi et al., 2015; Kang & McAuley, 2018) primarily focus on developing various temporal encoders to capture short- and long-term user interests. The evolution of temporal sequential encoders has progressed from LSTM units (Wu et al., 2017) and GRU units (Hidasi et al., 2015), to more advanced architectures such as graph neural networks (He et al., 2020; Xu et al., 2023b; 2024c; Zhao et al., 2023), self-attention layers (Kang & McAuley, 2018; Xu et al., 2024b), and Transformer models (Sun et al., 2019). Following the triumph of large language models (LLMs), researchers have begun leveraging open-source LLMs (Touvron et al., 2023) to construct their recommendation systems (Zhao et al., 2024; Bao et al., 2023; Wei et al., 2024). G-LLMRec methods (Geng et al., 2022; Xu et al., 2023a; Zhang et al., 2023b; Liao et al., 2023; Mei & Zhang, 2023) generate the next item based on historical sequences, while E-LLMRec approaches (Li et al., 2023a; Zhu et al., 2023; Wang et al., 2024) use LLMs as feature extractors to learn user representations for prediction. More recently, (Zhai et al., 2024) introduces a generative sequential framework scalable up to GPT-3 dimensions. LLM-based recommendation systems frequently outperform TSR models by a margin of 20% (Li et al., 2023a; Liao et al., 2023; Wang et al., 2024), also increasing the parameters by nearly 100 times compared to TSR models. Therefore, the deployment of LLMRec models in real-world platforms is heavily constrained by computational resources.

**Knowledge Distillation (KD).** Training a smaller "student" model on the distribution predicted by a large "teacher" model is known as a powerful knowledge distillation technique (Hinton et al., 2015). The fundamental insight behind this is to transform the knowledge and capabilities of the teacher into more compact, compressed, and possibly skill-specific representations (Jiao et al., 2020; Gu et al., 2024). For those cases when the student only has access to the output tokens generated by the teacher, another way of KD is data distillation (Eldan & Li, 2023; Li et al., 2023b; Fu et al., 2023; Hsieh et al., 2023). This technique first generates high-quality synthetic data by prompting the larger teacher model. The synthetic data are then used to enhance the student's capabilities by fine-tuning.

## 8  CONCLUSIONS

This paper explores the effectiveness of large language models (LLMs) in sequential recommendation. Our motivational experiments reveal that intermediate layers in LLMs are redundant for achieving optimal recommendation performance. Motivated by empirical insights, we adopt vanilla knowledge distillation methods to improve the performance of small language models. Achieving only 13% of the parameters compared to the LLMRec baseline, our SLMREC model yields an 8x acceleration and slightly better performance. On top of our technical contributions, we believe the results in this paper could shed light on a new promising direction for building effective and efficient recommenders based on LLMs, which is largely under-explored. Additionally, we provide theoretical justifications showing that while multi-layer models optimize distinct objectives at each layer, this can introduce redundancy compared to a single-layer model that achieves its objective in one step. These theoretical insights align with the motivation behind our framework design, where we employ knowledge distillation (KD) to guide the one-layer network, enabling it to streamline the learning process and replicate the feature extraction capabilities of a multi-layer network.

**Future Work.** This work concentrates on enhancing the efficiency of Large Language Model (LLM) utilization in the sequential recommendation. A notable limitation is the model's inability to adapt to new scenarios through few-shot learning. When confronted with a fresh dataset or new traffic logs from the platform, the model requires retraining from the entire dataset. In contrast, LLMs have demonstrated promising results in adapting to downstream language tasks using few-shot learning approaches. Looking ahead, we intend to investigate the incorporation of incremental learning into LLM-based recommendations to bolster the model's transferability. Additionally, integrating auxiliary linguistic and visual information of users and items into the LLMRec model may offer further improvements in its adaptability to new scenarios.

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

CONTENTS

## A    PROOF

### A.1    PROOF FOR PROPOSITION 1

**Proposition 1.** *Given the matrix* $\hat{\mathbf{A}}^{(k)} = \mathbf{A}^{(k)} + \mathbf{I}$, *Eqn. 7 is equivalent to a gradient descent step with step size 1 for the following optimization problem:*

$$\min_{H} \left\| \mathbf{H} - \hat{\mathbf{A}}^{(k)} \mathbf{H}^{(k-1)} \right\|_2^2 \tag{10}$$

*Proof.* The cost function at the $k$-th layer is denoted by

$$E(\mathbf{H}; \mathbf{H}^{(}k-1)) = \min_{\mathbf{H}} \left\| \mathbf{H} - \hat{\mathbf{A}}^{(k)} \mathbf{H}^{(k-1)} \right\|_2^2 \tag{11}$$

Then, the gradient of $E(\mathbf{H}; \mathbf{H}^{(}k-1))$ is computed as

$$\frac{\partial E(\mathbf{H}; \mathbf{H}^{(}k-1))}{\partial \mathbf{H}} = 2 \left( \mathbf{H} - \hat{\mathbf{A}}^{(k)} \mathbf{H}^{(k-1)} \right) \tag{12}$$

With the step size 1 of the gradient descent, it minimizes the cost function $E(\mathbf{H}; \mathbf{H}^{(}k-1))$ at the current layer is

$$\mathbf{H}^{(k)} = \mathbf{H}^{(k-1)} - \frac{\partial E(\mathbf{H}; \mathbf{H}^{(}k-1))}{\partial \mathbf{H}} \Big|_{\mathbf{H} = \mathbf{H}^{(k-1)}} \tag{13}$$

$$= \mathbf{H}^{(k-1)} - \left( \mathbf{H}^{(k-1)} - \hat{\mathbf{A}}^{(k)} \mathbf{H}^{(k-1)} \right) \tag{14}$$

$$= \hat{\mathbf{A}}^{(k)} \mathbf{H}^{(k-1)} \tag{15}$$

$$= \mathbf{H}^{(k-1)} + \mathbf{A}^{(k)} \mathbf{H}^{(k-1)} \tag{16}$$

$$\square$$

### A.2    PROOF FOR PROPOSITION 2

**Proposition 2.** *For any $K$-layer attention model (where $K$ is an arbitrary positive integer) with the layer-wise updating rule defined by Eqn. 7, there exists $\mathbf{C}^*$ such that one gradient descent step for the optimization problem (from the initial embeddings $\mathbf{H}^{(0)}$)*

$$\min_{\mathbf{H}} \left\| \mathbf{H} - \mathbf{C}^* \mathbf{H}^{(0)} \right\|_2^2, \tag{17}$$

*where $\mathbf{C}^*$ associated with $\mathbf{A}$, can yield the output embeddings $\mathbf{H}^{(K)}$ of the $K$-layer model.*

*Proof.* Similar to Theorem 1, we define $\hat{\mathbf{A}}^{(k)}$ to simpify the Eqn. 7.

$$\hat{\mathbf{A}}^{(k)} = \mathbf{I} + \mathbf{A}^{(k)}, \tag{18}$$

Then Eqn. 7 can be equivalently written as

$$\mathbf{H}^{(k)} = \hat{\mathbf{A}}^{(k)} \mathbf{H}^{(k-1)}, \tag{19}$$

By stacking $K$ layers of propagation, we can denote the output embeddings as

$$\mathbf{H}^{(K)} = \hat{\mathbf{A}}^{(K)} \mathbf{H}^{(K-1)} = \hat{\mathbf{A}}^{(K)} \hat{\mathbf{A}}^{(K-1)} \mathbf{H}^{(K-2)} = \cdots = \hat{\mathbf{A}}^{(K)} \cdots \hat{\mathbf{A}}^{(1)} \mathbf{H}^{(0)} = \mathbf{A}^* \mathbf{H}^{(0)}, \tag{20}$$

where $\mathbf{A}^*$ defined as multiple matrix production.

We can show that solving the denoising problem with gradient step size $\frac{\mu^*}{2}$ w.r.t. the objective

$$\min_{\mathbf{H}} \left\| \mathbf{H} - \mathbf{C}^* \mathbf{H}^{(0)} \right\|_2^2, \tag{21}$$

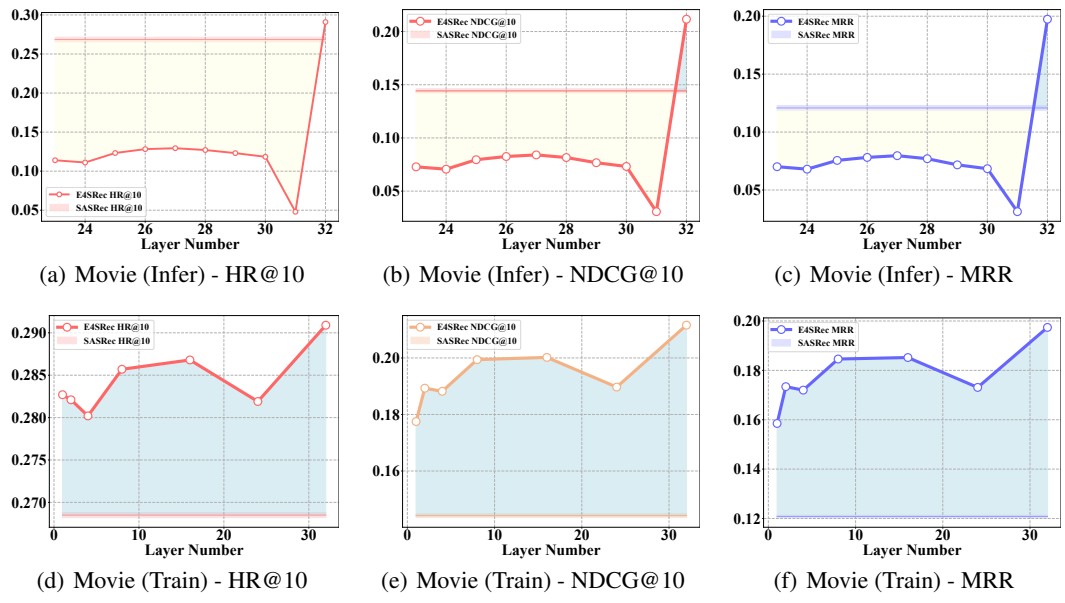

Figure 5: We present the relationship between the number of decoder layers and the final recommendation performance, with the performance of SASRec plotted as a baseline. Figures (a)-(c) show the results of directly using representations from the middle layers for inference without training, while (d)-(f) prune the later layers and train a model using only the specified number of layers. From the results, we observe that deeper decoder layers introduce redundancy in recommendation tasks, with models utilizing fewer layers (8-layer) achieving performance nearly equivalent to (24-layer) models.

Defining $\mathbf{C}^* = \frac{1}{\mu^*} \left( \mathbf{A}^* - (1 - \mu^*)\mathbf{I} \right)$, $\mathbf{H}^{(k)} = \mathbf{A}^*\mathbf{H}^{(0)}$ will induce the output embeddings $\mathbf{H}^{(K)}$, by noticing that

$$\mathbf{H}^{(K)} = \mathbf{H}^{(0)} - \frac{\mu^*}{2} \left. \frac{\partial E(\mathbf{H}; \mathbf{H}^{(0)})}{\partial \mathbf{H}} \right|_{\mathbf{H}=\mathbf{H}^{(0)}} \tag{22}$$

$$= \mathbf{H}^{(0)} - 2\frac{\mu^*}{2}(\mathbf{H}^{(0)} - \mathbf{C}^*\mathbf{H}^{(0)}) \tag{23}$$

$$= \mathbf{H}^{(0)} - 2\frac{\mu^*}{2}[\mathbf{H}^{(0)} - \frac{1}{\mu^*}\left( \mathbf{A}^* - (1 - \mu^*)\mathbf{I} \right)\mathbf{H}^{(0)}] \tag{24}$$

$$= \mathbf{H}^{(0)} - \mu^*\mathbf{H}^{(0)} + \mathbf{A}^*\mathbf{H}^{(0)} - \mathbf{H}^{(0)} + \mu^*\mathbf{H}^{(0)} \tag{25}$$

$$= \mathbf{A}^*\mathbf{H}^{(0)} \tag{26}$$

$\square$

# B EXPERIMENTS

## B.1 MOTIVATION EXPERIMENT RESULTS

## B.2 TRAINING DETAILS

In Table 6, we provide hyper-parameters in our training stage. Our implementation is based on Huggingface Transformers [6]. The input and intermediate hidden dimension in the feed-forward network is 4096. We use mixed precision training and train on 1*80G Nvidia A100 GPU.

## B.3 COMPARED METHODS

**Tranadtional sequential recommendation methods:**

---

[6] https://github.com/huggingface/transformers

Table 6: Hyper-parameter (HP) settings of our method on each dataset.

| HP | Cloth | Movie | Music | Sport |
|---|---|---|---|---|
| adam_beta1 | 0.9 | 0.9 | 0.9 | 0.9 |
| adam_beta2 | 0.999 | 0.999 | 0.999 | 9.999 |
| adam_epsilon | 1e-8 | 1e-8 | 1e-8 | 1e-8 |
| learning_rate | 0.003 | 0.001 | 0.002 | 0.002 |
| logging_steps | 1 | 1 | 1 | 1 |
| lr_scheduler_type | cosine | cosine | cosine | cosine |
| max_grad_norm | 1.0 | 1.0 | 1.0 | 1.0 |
| max_steps | 1500 | -1 | 800 | 2000 |
| optimizer | adamw_torch | adamw_torch | adamw_torch | adamw_torch |
| save_strategy | steps | steps | steps | steps |
| save_steps | 50 | 100 | 100 | 100 |
| eval_steps | 50 | 100 | 100 | 100 |
| warmup_steps | 50 | 50 | 100 | 50 |
| $\lambda_1$ | 1.0 | 1.0 | 1.0 | 1.0 |
| $\lambda_2$ | 0.1 | 0.1 | 0.1 | 0.1 |
| $\lambda_3$ | 1.0 | 1.0 | 0.01 | 0.1 |
| $b$ | 4 | 4 | 4 | 4 |

**Caser** (Tang & Wang, 2018) introduces a novel approach to sequential recommendation systems by modeling user-item interactions as sequences, which is designed to predict the next item a user may interact with by capturing both short-term and long-term dependencies in user behavior.

**GRU4Rec** (Hidasi et al., 2015) tackles the issue of modeling sparse sequential data while also adapting RNN models to the recommender system. To achieve this, the authors propose a new ranking loss function that is specifically designed for training these models. The implementation of GRU4Rec in PyTorch can be found at the URL [7].

**BERT4Rec** (Sun et al., 2019) designs a bidirectional self-attention network to model user behavior sequences. To prevent information leakage and optimize the training of the bidirectional model, a Cloze objective is used to predict the randomly masked items in the sequence by considering both their left and right context. The implementation of BERT4Rec in PyTorch can be found at the URL [8].

**SASRec** (Kang & McAuley, 2018) is a self-attention based sequential model that addresses the challenge of balancing model parsimony and complexity in recommendation systems. By using an attention mechanism, SASRec identifies relevant items in a user's action history and predicts the next item based on relatively few actions, while also capturing long-term semantics like an RNN. This enables SASRec to perform well in both extremely sparse and denser datasets. The implementation of SASRec in PyTorch can be found at the URL [9].

**HGN** (Ma et al., 2019) propose a novel hierarchical gating mechanism to effectively capture both short-term and long-term user preferences in sequential recommendation tasks. Their model dynamically selects relevant interaction history at multiple temporal levels, improving next-item prediction accuracy. This approach outperforms state-of-the-art methods while maintaining efficiency and scalability for large-scale recommendation systems.

**LightSANs** (Fan et al., 2021) introduces a low-rank decomposition technique for self-attention networks, reducing their computational complexity while maintaining strong performance. This approach makes the model more efficient and scalable for large-scale recommendation tasks without compromising accuracy.

For the code implementation of Caser, HGN and LightSANs, we run the experiment based on the RecBole (Zhao et al., 2021) [10].

---

[7] https://github.com/hungpthanh/GRU4REC-pytorch
[8] https://github.com/jaywonchung/BERT4Rec-VAE-Pytorch
[9] https://github.com/pmixer/SASRec.pytorch
[10] https://github.com/RUCAIBox/RecBole/tree/master

**Self-supervised sequential recommendation methods**

**S³-Rec** (Zhou et al., 2020) introduces self-supervised learning into a sequential recommendation by utilizing mutual information maximization (MIM) to learn better representations from user sequence data. The model incorporates four auxiliary self-supervised objectives: item cropping, item masking, item reordering, and segment prediction, to enhance the quality of learned item representations. By pre-training the model with these self-supervised tasks and then fine-tuning in the recommendation task, it achieves strong performance even with limited training data. The method demonstrates that self-supervised learning can effectively leverage the inherent supervisory signals within sequential data to improve recommendation quality.

**DuoRec** (Qiu et al., 2022) addresses the representation degeneration problem in sequential recommendation where item embeddings tend to be similar and occupy a narrow cone. Instead of traditional data-level augmentation (like masking or cropping), it introduces model-level augmentation using different Dropout masks to generate sequence representations. They also propose using sequences with the same target item as positive samples, which provides more meaningful contrasts. Through this contrastive regularization approach, DuoRec encourages a more uniform distribution of embeddings in the representation space, leading to better recommendation performance. The key innovation is tackling representation degeneration through model architecture rather than data augmentation while maintaining semantic consistency in the contrasting process.

**MAERec** (Ye et al., 2023) addresses limitations in sequential recommendation systems where other methods struggle with limited labels and noisy user behavior. Unlike previous approaches using manual contrastive learning strategies, MAERec implements a graph masked autoencoder framework that automatically identifies and focuses on meaningful item relationships. The model uses adaptive masking and task-specific regularization to ensure the learned representations align with recommendation goals while filtering out noise. By dynamically reconstructing masked item transitions rather than relying on hand-crafted data augmentation, MAERec achieves more robust performance across different recommendation scenarios without requiring manual heuristics. The approach demonstrates strong results in handling both data sparsity and noise while maintaining computational efficiency. The implementation of MAERec in PyTorch can be found at the URL [11].

For the code implementation of S³-Rec and DuoRec, we run the experiment based on the RecBole (Zhao et al., 2021) [12] [13].

**LLM-based recommendation methods:**

**Open-P5** (Xu et al., 2023a) is an open-source platform introduced to catalyze research in LLM-based generative recommender systems. It supports key model architectures like T5 and Llama-2 across diverse public datasets, focusing on sequential and straightforward recommendation tasks. The platform emphasizes the role of item IDs through various indexing methods and offers a customizable, efficient, and standardized environment for developing and assessing recommender systems. The implementation of Open-P5 in PyTorch can be found at the URL [14].

**E4SRec** (Li et al., 2023a) integrate of Large Language Models (LLMs) into sequential recommendation systems, offering a significant leap in handling item IDs and personalization. In the original paper, they use Softmax layer to output each user-item prediction score. The implementation of E4SRec in PyTorch can be found at the URL [15].

### B.4 MODEL EFFICIENCY

We show the running time of Open-P5, E4SRec, and our SLMREC in each dataset. These comparisons were conducted on a machine with an A100 GPU. The training batch size for all models was standardized at 256. During inference, E4SRec and SLMREC utilized a batch size of 512, whereas Open-P5's inference was performed with a batch size of 1.

---

[11] https://github.com/HKUDS/MAERec/tree/main
[12] https://github.com/RUCAIBox/RecBole/tree/master
[13] https://github.com/RuihongQiu/DuoRec/tree/master
[14] https://github.com/agiresearch/OpenP5
[15] https://github.com/HestiaSky/E4SRec

Table 7: Detailed efficiency comparison of Open-P5, E4SRec, and our SLMREC, in terms of training and inference time, on each dataset.

| Method | Cloth | | Movie | | Music | | Sport | |
|---|---|---|---|---|---|---|---|---|
| | Tr time (h) | Inf time (h) | Tr time (h) | Inf time (h) | Tr time (h) | Inf time (h) | Tr time (h) | Inf time (h) |
| Open-P5$_{\text{LLaMa}}$ | 1.36 | 3554.43 | 0.36 | 3504 | 0.35 | 3692 | 1.60 | 9017 |
| E4SRec | 5.27 | 0.578 | 1.90 | 0.208 | 1.88 | 0.216 | 6.75 | 0.660 |
| **SLMREC**$_{4\leftarrow8}$ | 0.97 | 0.070 | 0.15 | 0.030 | 0.30 | 0.030 | 0.98 | 0.078 |

Table 8: Efficiency comparison of SASRec, MAERec and our SLMREC in terms of epoch-wise inference time (hours). These comparisons were conducted on a machine with an A100 GPU. During inference, models leverage parallel processing with a batch size of 512.

| Method | Inf time(h) | Improv. (%) |
|---|---|---|
| SASRec | 0.015 | 0.00 |
| MAERec | 0.061 | 11.96 |
| **SLMREC**$_{4\leftarrow8}$ | 0.052 | 45.26 |

Table 9: Experiment results (%) of ablation study.

| SLMREC | Music | | | | Sport | | | |
|---|---|---|---|---|---|---|---|---|
| | **HR@1** | **HR@5** | **NDCG@5** | **MRR** | **HR@1** | **HR@5** | **NDCG@5** | **MRR** |
| +$\mathcal{D}_{cos}$ | 5.62 | 8.78 | 7.23 | 7.81 | 6.25 | 9.25 | 7.76 | 8.41 |
| +$\mathcal{D}_{cos},\mathcal{D}_{norm}$ | **5.95** | **9.26** | **7.65** | **8.23** | 6.61 | 9.82 | 8.24 | 8.87 |
| +$\mathcal{D}_{cos},\mathcal{L}_{ms}$ | 5.69 | 8.94 | 7.36 | 7.91 | 6.51 | 9.39 | 7.96 | 8.62 |
| +$\mathcal{D}_{cos},\mathcal{D}_{norm},\mathcal{L}_{ms}$ | 5.72 | 9.15 | 7.48 | 8.03 | **6.62** | **9.83** | **8.25** | **8.89** |

To evaluate deployment efficiency in real-world scenarios, we compare the inference time between traditional recommendation approaches and LLM-based methods. Our experimental results demonstrate that SLMRec achieves comparable computational efficiency to traditional methods while delivering a substantial performance improvement of nearly 50% over SASRec.

## B.5 ABLATION STUDY

We present the remaining ablation study results in Table 9. SLMREC, when enhanced with various knowledge regularizers (namely $\mathcal{D}_{cos}, \mathcal{D}_{norm}$ and $\mathcal{L}_{ms}$), demonstrates improved performance. The regularizers $\mathcal{D}_{cos}$ and $\mathcal{D}_{norm}$ aid SLMREC in aligning its intermediate representations with those of the teacher model, thereby endowing it with more potent representational extraction capabilities. Meanwhile, $\mathcal{L}_{ms}$ steers the model to assimilate domain knowledge pertinent to recommendation systems within its preliminary layers.

## C  EXTENDED RELATED WORK

### C.1  DEPTH-WISE KNOWLEDGE OF LLMS

The recent community interest stems from how linguistic properties and knowledge are encoded in language models. (Meng et al., 2022; Dai et al., 2022; Jin et al., 2025) emphasize that knowledge localizes within the middle or final layers. On the other hand, (Hase et al., 2024) attempts to perform knowledge editing and concludes that information may be stored non-locally across layers. What's more, (Men et al., 2024; Gromov et al., 2024) share a similar view that current pretraining methods are not properly leveraging the parameters in the deeper layers of the network or that the shallow layers play a critical role in storing knowledge. By contrast, we are the first to investigate which part of knowledge on the LLMs plays a key role, especially in the sequential recommendation scene.

### C.2  MODEL PRUNING

Model Pruning is a fundamental approach for reducing the size of a well-trained large model by removing unimportant parameters (Hassibi & Stork, 1992). Recent work has focused on applying pruning methods to the Transformer architecture (Vaswani et al., 2017). These works have studied different components of the model architecture for pruning, including dropping attention heads (Voita et al., 2019; Michel et al., 2019), dropping layers (Fan et al., 2019; Zhang & He, 2020; Kim & Awadalla, 2020; Sajjad et al., 2023), dropping hidden xistates (Hou et al., 2020), replacing sparse weight matrices with smaller dense ones (Ashkboos et al., 2024), and combinations of these solutions. By contrast, our work performs layer removal through simple knowledge distillation, rather than more complex pruning techniques.

### C.3  PARAMETER-EFFICIENT FINE-TUNING (PEFT)

PEFT emerges as a novel technique for tailoring Large Language Models (LLMs) to specific tasks while ensuring minimal computational and memory costs (Houlsby et al., 2019; Lester et al., 2021; Hu et al., 2021b; Liu et al., 2022). In this work, we combine our method with the Low-Rank Adapters (LoRA) (Hu et al., 2021b) to reduce the memory and computation of the knowledge distillation process. Specifically, we freeze the pre-trained model and only tune a small set of additional trainable parameters.

