# OpenReview forum: "SLMRec: Distilling Large Language Models into Small for Sequential Recommendation"
_ICLR.cc/2025/Conference — ICLR 2025 Poster_

### Official Review · Reviewer_ZHQw · 2024-10-17

**Soundness:** 2
**Presentation:** 3
**Contribution:** 2
**Rating:** 6
**Confidence:** 4

**Summary:**

This paper works on improving the efficiency of LLM as a recommender scheme by arguing that the intermediate layer of LLM is redundant. Specifically, the paper suggests a vallina knowledge distillation approach to achieve near LLM-based Recommendations in an efficient way.

**Strengths:**

1.	The paper addresses an important problem regarding the efficiency of LLM as a recommender scheme and provides a reasonable knowledge distillation approach to solving the problem.
2.	The paper is generally well-written.

**Weaknesses:**

1.	The knowledge distillation approach is widely used in LLM-related work [1], which raises concerns about its novelty when adopting it in the recommendation.
2.	Although the paper claims that they provide both experimental and theoretical justification for adopting KD to LLMRec. It does not look strong enough.
    1. In the experimental part, as only 8-layer and 24-layer are discussed, could the author provide a more detailed analysis of the performance differences between the 8-layer, 24-layer, and 32-layer models?
    2. In terms of the theoretical part, the author suggested that a small model can also learn user representation that is similar to the larger model, while it does not align with the recent work on the emergent ability of LLM that a small model cannot achieve a similar result of the large language model.
Could the author discuss how your findings relate to existing work on emergent abilities in LLMs, and explain why user representation learning may differ from other tasks where emergent abilities have been observed? Also, some supportive experiment is needed to justify this claim.


[1] Xu X, Li M, Tao C, Shen T, Cheng R, Li J, Xu C, Tao D, Zhou T. A survey on knowledge distillation of large language models. arXiv preprint arXiv:2402.13116. 2024 Feb 20.

**Questions:**

Could the author justify the claim that “an 8-layer E4SRec can obtain nearly as informative user representation as a 24-layer E4SRec” by answering the following question in detail?
1. Provide a more detailed analysis of the performance differences between the 8-layer, 24-layer, and 32-layer models.
2. Explain the apparent performance surge at 32 layers and discuss its implications for your claims.

---

> ### Author Response · Authors · 2024-11-20
> **Response to Reviewer ZHQw (Part 1 of 2)**
>
> We truly appreciate your constructive criticism and suggestions, which have helped us refine and strengthen the paper. We aim to clarify any misunderstandings about our technical contributions by addressing your questions point by point.
>
> > ***"Q1. Novelty of knowledge distillation for LLM."***
>
> **Distinctions in knowledge distillation.** Recent LLM knowledge distillation methods, as illustrated in Fig. 1 of [1], focus on transferring knowledge from closed-source teacher models (such as GPT-4 and Claude) to improve smaller open-source models (like LLaMA). Due to the closed-source characteristic of these teacher models, researchers can only access their generated results, not their intermediate representations.
>
> In contrast, our SLMRec takes a different approach. We utilize the hidden representations from our open-source teacher model to guide the student model in developing similar feature extraction capabilities. Our teacher and student models are open-source and specifically trained on recommendation tasks, making them domain-specialized models. This technical distinction sets our proposed method apart from recent LLM knowledge distillation approaches [1].
>
> **Distinctions in motivation.** The knowledge distillation approaches in [1] aim to leverage advanced capabilities from proprietary models like GPT-4 or Gemini to enhance open-source LLMs. They aim at distilling high-quality data from the closed-source LLMs
>
> In contrast, our SLMRec's motivation stems from our motivational experiments detailed in Section 2. Based on these experimental insights, we implement feature distillation to guide the student model in learning to encode hidden representations similar to those of the teacher model.
>
> **Efficiency.** Experimental results show that SLMRec achieves state-of-the-art performance using just 13% of the parameters of current LLM-based recommendation models while providing substantial computational benefits: 6.6x faster training and 8.0x faster inference times. Based on Google Colab's pricing for A100 GPU instances, our algorithm can reduce model serving costs by approximately $1,000 per month per A100 GPU. From a practical perspective, our SLMRec offers insights into efficiently deploying lightweight language models for recommendation systems.
>
> > ***"Q2. Explanation of the claim in the motivational experiment."***
>
> Thank you for your interesting question. We have discussed the phenomenon in the initially submitted PDF.
>
> **Explain about 32-layer model performance.** When the number of layers increases, the performance of the model also improves. We specifically focus on the 8-24 layer range because the performance improvement in this range is slight. While we observe a more significant performance gain from 24 to 32 layers, we hypothesize this is due to the complete 32-layer architecture being utilized during pre-training despite not being pre-trained specifically for recommendation tasks.
>
> **Our findings.** Our motivational experiments are based on empirical observations, with additional supporting evidence in the appendix showing consistent patterns. When varying the number of layers from 8 to 24, we observe only marginal performance improvements. Specifically, an 8-layer E4SRec can generate user representations nearly as informative as those from a 24-layer E4SRec. This observation suggests that intermediate layers in LLMRec models may be redundant, and smaller language models may potentially match the feature extraction capabilities of larger models through effective knowledge transfer.

---

> ### Author Response · Authors · 2024-11-20
> **Response to Reviewer ZHQw (Part 2 of 2)**
>
> > ***"Q3. Explanation of the theoretical part."***
>
> **Emergent abilities.** Emergent abilities show how model capabilities suddenly appear at certain parameter scales [2]. All the experiments and insights are designed in the NLP fields. All LLMs employed in our study are trained and optimized for recommendation tasks, serving as domain-specific recommendation models. Meanwhile, the phenomenon of emergent abilities within the recommendation domain remains largely unexplored and presents an open research question. Therefore, there is no conflict between their conclusions and our research findings.
>
> Recent works [3-6] have focused on aligning LLMs with human preferences and implementing data-cleaning mechanisms to improve small language model performance. Their findings show that well-curated data allows smaller models to achieve comparable results to larger ones, paralleling the core principle of our SLMRec approach.
>
> **Our theoretical justification.** Our theoretical justification indicates that for any multi-layer stacked decoder, an equivalent single-layer decoder can be constructed to encode hidden representations in a similar way. Moreover, while the multi-layer model optimizes distinct objectives at each layer, this may introduce redundancy compared to a single-layer model that achieves its objective in a single step.
>
>
> Reference:
>
> [1] Xu, Xiaohan, et al. "A survey on knowledge distillation of large language models." arXiv preprint arXiv:2402.13116 (2024).
>
> [2] Wei, Jason, et al. "Emergent abilities of large language models." arXiv preprint arXiv:2206.07682 (2022).
>
> [3] Chen, Zixiang, et al. "Self-play fine-tuning converts weak language models to strong language models." ICML 2024.
>
> [4] Bansal, Hritik, et al. "Smaller, weaker, yet better: Training llm reasoners via compute-optimal sampling." arXiv preprint arXiv:2408.16737 (2024).
>
> [5] Rosset, Corby, et al. "Direct nash optimization: Teaching language models to self-improve with general preferences." arXiv preprint arXiv:2404.03715 (2024).
>
> [6] Wu, Yue, et al. "Self-play preference optimization for language model alignment." arXiv preprint arXiv:2405.00675 (2024).

---

> ### Author Response · Authors · 2024-11-26
> **Kindly Reminder**
>
> Dear Reviewer,
>
> We have submitted our detailed rebuttal addressing all raised concerns. We would greatly appreciate it if you could review our response and let us know if any aspects require further clarification.
>
> Best regards,
>
> Authors

---

> > ### Comment · Reviewer_ZHQw · 2024-11-26
> >
> > Thanks for the author's reply.
> > For Q1, thank you for justifying the distinction between recent LLM knowledge distillation and the proposed approach, while the novelty concern remains.
> >
> > For Q2, the question remains, first, you stated that 32 layers have better performance, meanwhile, you argue that the intermediate layers in LLMRec models may be redundant as models with layers from 8 to 24 have similar capabilities. Making it hard to understand why the argument in the paper "deeper decoder layers introduce redundancy in recommendation tasks" holds. Also, your hypothesis on: "we hypothesize this is due to the complete 32-layer architecture being utilized during pre-training despite not being pre-trained specifically for recommendation tasks." is hard to understand, why 32-layer architecture which is utilized during pretraining makes it performs better than 24 layers architecture (which is also utilized during pretraining) when the parameters are not modified.
> >
> > For Q3, Thank you for addressing the concern, the discussion on emergent ability makes sense to me.

---

> > > ### Author Response · Authors · 2024-11-26
> > > **Response to Reviewer ZHQw (Part 1 of 2) - Addressing the Rested Concerns**
> > >
> > > Thank you for your constructive feedback. We are pleased that we could effectively **address your concern about our theoretical part**. For the rest of the concerns regarding novelty and empirical motivational experiments, we have provided comprehensive responses to each point raised.
> > >
> > > > ***"Q1. The novelty of our work."***
> > >
> > > Novelty is mainly based on subjective evaluation, but we'd like to argue for our technical contributions that can be justified by illuminating the differences from existing works.
> > >
> > > We summarize the novelty of our method into three contributions.
> > >
> > > **Contribution 1: Empirical Insights.** Through extensive experiments on large-scale Amazon-18 datasets (containing over 10 million user-item interactions), we discover that LLMs exhibit significant redundancy in recommendation tasks, as models with 8-24 layers achieve comparable performance to deeper architectures. This empirical finding challenges the conventional wisdom that larger models are necessarily better, motivating the optimization of LLM-based recommendation models for improved efficiency.
> > >
> > > **Contribution 2: A general knowledge distillation framework.** We empower small language models for sequential recommendation with a general knowledge distillation framework. We adopt the vanilla knowledge distillation approaches to align the representation knowledge. It can equipped with all existing LLM-based recommendation models and
> > > requires no modifications to the internal architecture.
> > >
> > > **Contribution 3: Theoretical Justification for Model Efficiency.** We provide rigorous theoretical analysis demonstrating the effectiveness of smaller models through two key propositions: (1) establishing the equivalence between multi-layer attention mechanisms and gradient-based optimization dynamics and (2) proving the existence of an equivalent single-layer representation that can achieve comparable expressiveness to multi-layer architectures. This theoretical framework mathematically validates that properly designed smaller models can achieve competitive performance while significantly reducing model complexity.
> > >
> > > This work presents a comprehensive solution to the efficiency challenges in LLM-based recommendation systems, supported by both empirical evidence and theoretical guarantees.
> > >
> > > We sincerely hope the above clarification helps to address the reviewer's lingering concern about the novelty. If not, we'd appreciate any detailed justification or related literature that weakens the contributions of our work.

---

> > > ### Author Response · Authors · 2024-11-26
> > > **Response to Reviewer ZHQw (Part 2 of 2) - Addressing the Rested Concerns**
> > >
> > > > ***"Q2. Explanation about the motivational experiments."***
> > >
> > > Thank you for raising this point.
> > >
> > > **On model redundancy.** Recent studies in NLP tasks have provided important insights into layer-wise model behaviors.
> > >
> > > Analyses of LLM architectures [1] reveal that while deeper layers generally exhibit similar representational patterns, the final layers demonstrate distinctive behaviors with maximal dissimilarity from earlier layers, suggesting their specialized role in task-specific adaptation.
> > >
> > > Similarly, investigations of knowledge encoding [2] show that intermediate layers primarily process basic knowledge, while final layers are crucial for context integration and knowledge synthesis. These findings strongly align with our observations in recommendation tasks - the redundancy we observe in layers 8-24 mirrors these established patterns.
> > >
> > > These observations align with our findings and motivate our knowledge distillation approach to efficiently preserve model capabilities while reducing redundancy.
> > >
> > > **On our hypothesis.** The progressive nature of feature improvement in deep neural networks has been well-documented in computer vision studies [3][4]. Our observations in LLM-based recommendation align with these findings: while increasing the number of layers leads to better performance, this improvement follows a gradual pattern rather than linear gains.
> > >
> > > Also, the hypothesis aligns with the phenomena in NLP tasks: (1) preserving upper layers is crucial for maintaining abstract reasoning capabilities from pre-training [5], and (2) the full architecture enables optimal utilization of different layer combinations for specific tasks [6].
> > >
> > > This explains our empirical observation that model performance improves with additional layers, but in a progressive manner where each layer builds upon and refines the representations of previous layers rather than providing dramatic individual improvements.
> > >
> > > We hope this explanation helps address the questions raised about layer dynamics. We appreciate the thoughtful feedback and welcome further discussion that could deepen our understanding of these observations.
> > >
> > > Reference:
> > >
> > > [1] Gromov, Andrey, et al. "The unreasonable ineffectiveness of the deeper layers." arXiv preprint arXiv:2403.17887 (2024).
> > >
> > > [2] Ju, Tianjie, et al. "How large language models encode context knowledge? a layer-wise probing study." arXiv preprint arXiv:2402.16061 (2024).
> > >
> > > [3] He, Kaiming, et al. "Deep residual learning for image recognition." Proceedings of the IEEE conference on computer vision and pattern recognition. 2016.
> > >
> > > [4] Karras, Tero, et al. "Analyzing and improving the image quality of stylegan." Proceedings of the IEEE/CVF conference on computer vision and pattern recognition. 2020.
> > >
> > > [5] Clark, Kevin. "What Does Bert Look At? An Analysis of Bert’s Attention." arXiv preprint arXiv:1906.04341 (2019).
> > >
> > > [6] Liu, Liyuan, et al. "Understanding the difficulty of training transformers." arXiv preprint arXiv:2004.08249 (2020).

---

> > > > ### Comment · Reviewer_ZHQw · 2024-11-27
> > > >
> > > > Thanks for the explanations. The study on knowledge encoding make sense to me.

---

> > > > > ### Author Response · Authors · 2024-11-27
> > > > >
> > > > > Thank you for your thoughtful comments and raising the score! Your feedback and subsequent discussions have significantly strengthened the paper's quality.

---

### Official Review · Reviewer_M71Q · 2024-11-03

**Soundness:** 3
**Presentation:** 3
**Contribution:** 3
**Rating:** 6
**Confidence:** 3

**Summary:**

This work investigates the use of small language models (SLMs) in the realm of sequential recommendation systems, proposing a model called SLMRec. The authors identify that traditional sequential recommendation (TSR) methods have reached a performance plateau and that existing large language model (LLM) approaches often require significant computational resources due to their large size. The proposed SLMRec aims to maintain competitive performance with a much smaller parameter footprint (less than 1 billion parameters) compared to the over 7 billion parameters typical of LLMs used in recommendations. The authors utilize knowledge distillation techniques to align the representation knowledge between a larger teacher model and the smaller student model, achieving an up to 8x acceleration in both training and inference times.

**Strengths:**

S1. SLMRec demonstrates significant efficiency improvements, achieving competitive performance while using substantially fewer parameters than larger models.
S2. The model's design allows it to operate within the resource constraints typical of real-world applications, making it more applicable for widespread use in industry.
S3. The paper's approach to knowledge distillation, focusing on feature alignment rather than just output prediction, is a noteworthy contribution that can lead to better representation learning in smaller models.
S4. The authors conduct comprehensive experiments on large-scale datasets, providing empirical evidence of the model's effectiveness across multiple metrics and domains.

**Weaknesses:**

W1. The implementation of knowledge distillation can introduce complexity in the training process, which may not be straightforward for all practitioners to apply.

**Questions:**

Please refer to the Weaknesses.

---

> ### Author Response · Authors · 2024-11-20
> **Response to Reviewer M71Q**
>
> Thank you for your thorough and insightful review of our work. We appreciate your recognition of SLMRec's significant efficiency improvements while maintaining competitive performance, its practical applicability within real-world resource constraints, our novel approach to knowledge distillation for enhanced representation learning, and our comprehensive experimental validation across multiple domains. Your detailed feedback highlights the broad impact and technical contributions of our research.
>
> > ***"Q1. The implementation of knowledge distillation is too complex to apply."***
>
> Our methodology primarily employs offline knowledge distillation, wherein the teacher model undergoes pre-training before the distillation process. As illustrated in Figure 4 and extensively discussed in Section 5.3, we also investigate the efficacy of online knowledge distillation in SLMRec, which eliminates the need for teacher model pre-training.
>
> For practitioners interested in implementing our framework with reduced computational complexity, we recommend exploring the online knowledge distillation variant. However, it is crucial to understand the inherent trade-offs: offline knowledge distillation typically ensures training stability and consistent performance, while online knowledge distillation offers implementation simplicity but may exhibit convergence variability across different scenarios [1]. Considering our objective of developing a robust and generalizable methodology across diverse datasets, we ultimately selected the offline knowledge distillation approach as our primary implementation strategy. Improving the stability of online knowledge distillation strategies represents a promising direction for future SLMRec development.
>
> Reference:
>
> [1] Gou, Jianping, et al. "Knowledge distillation: A survey." International Journal of Computer Vision 129.6 (2021): 1789-1819.

---

> ### Author Response · Authors · 2024-11-26
> **Kindly Reminder**
>
> Dear Reviewer,
>
> We have submitted our detailed rebuttal addressing all raised concerns. We would greatly appreciate it if you could review our response and let us know if any aspects require further clarification.
>
> Best regards,
>
> Authors

---

### Official Review · Reviewer_jtBY · 2024-11-03

**Soundness:** 3
**Presentation:** 3
**Contribution:** 3
**Rating:** 5
**Confidence:** 5

**Summary:**

This work discusses the impact of LLMs in SR, but questions the necessity of their size. The inefficiency of applying LLMs to real-world platforms due to their size is noted. The experiments show that most intermediate layers of LLMs are redundant, leading to the development of a small language model for SR using knowledge distillation. SLMRec achieves strong performance with significantly fewer parameters than LLM-based models, along with notable speedups in training and inference times. It also provides theoretical reasoning for the effectiveness of small language models in SR.

**Strengths:**

1. The authors explored the relationship between the parameter count of pre-trained LLMs and recommendation performance under the E-LLMRec paradigm, discovering a balance point between performance and parameter count that enhances the practicality of deployment in real-world scenarios.

2. The authors conducted thorough theoretical justifications to demonstrate why small language models can possess learning capabilities for user representations that are comparable to those of large language models.

3. The paper is well-written, and the figures clearly illustrate the overall workflow of SLMRec.

**Weaknesses:**

1. The motivation is not sufficiently clear. The authors claim that LLMs with fewer layers can achieve recommendation performance comparable to those with more layers, and thus they opt to use large LLMs to distill a small LLM. However, it is unclear why small LLMs cannot be fine-tuned directly. The authors also do not present results in the ablation study showing the fine-tuning of a 4-layer small LLM with CE loss without any distillation operation.

2. Generally speaking, distillation is performed to transfer knowledge from one model to another. However, small LLMs possess certain language capabilities on their own. During the distillation process, how can the teacher model be guided to transfer recommendation-related information to the student model? If only general knowledge is distilled, the parameter limitations of small LLMs imply that they may not be able to learn additional knowledge effectively.

3. Although knowledge distillation can help small LLMs acquire certain recommendation capabilities and reduce deployment costs, simply reducing the number of layers from 8 to 4, while being more efficient than Open-P5 and E4SRec, still renders it too inefficient compared to traditional sequence recommendation methods. This creates a significant gap in terms of practical deployment, weakening the motivation claimed by the authors.

4. The experimental section lacks some of the latest traditional sequence recommendation methods, such as self-supervised approaches (e.g., DuoRec [1], MAERec [2]) and pre-trained methods (e.g., S3-Rec [3]).

5. In Table 2, the experimental results on the Cloth dataset show that E4SRec outperforms SLMRec in HR@1, NDCG@5, and MRR. However, it is marked in purple as having "the second best performance," which seems inconsistent.

[1] Qiu, Ruihong, et al. "Contrastive learning for representation degeneration problem in sequential recommendation." Proceedings of the fifteenth ACM international conference on web search and data mining. 2022.

[2] Ye, Yaowen, Lianghao Xia, and Chao Huang. "Graph masked autoencoder for sequential recommendation." Proceedings of the 46th International ACM SIGIR Conference on Research and Development in Information Retrieval. 2023.

[3] Zhou, Kun, et al. "S3-rec: Self-supervised learning for sequential recommendation with mutual information maximization." Proceedings of the 29th ACM international conference on information & knowledge management. 2020.

**Questions:**

1. What are the results of fine-tuning small LLMs using only CE loss? (i.e., without employing knowledge distillation methods). Additionally, what would the results be if only both CE loss and MS loss were used? (It appears that MS loss is unrelated to knowledge distillation.)

2. Could the authors include comparisons with other traditional sequence recommendation methods in the Efficiency Comparison section to provide a more comprehensive evaluation of their efficiency?

3. Could the authors provide experimental results for the self-supervised and pre-trained sequence recommendation methods mentioned in the fourth point of the Weakness section?

---

> ### Author Response · Authors · 2024-11-20
> **Response to Reviewer jtBY (Part 1 of 2)**
>
> We truly appreciate your constructive feedback and suggestions, which have helped us strengthen the paper. In the following, we clarify the misunderstanding and supplement more comparison with three suggested baselines.
>
> > ***"Q1. What is the motivation for distilling a small language model rather than finetuning this model? Moreover, what are the experiment results without distillation?"***
>
> **Motivation.** In our motivational experiments, we utilize E4SRec [1] to investigate potential parameter redundancy in LLMs. Notably, both E4SRec and SLMRec enhance prediction efficiency by adopting a BERT-style prediction head [2] for recommendation tasks, rather than generating next-item predictions autoregressively. This approach positions LLMs in embedding-based methods, including Lite-LLM4Rec [3], CLLM4Rec [4], E4SRec, and our SLMRec, primarily as sequential feature extractors.
>
> Our experiments reveal that while model performance improves with increasing layer depth, the gains become marginal when scaling from 8 to 24 layers. This observed parameter redundancy motivates our adoption of knowledge distillation, enabling the student model to learn effective feature extraction strategies from the teacher model, as directly fine-tuning small language models may fail to achieve global optimization.
>
> **Experiment Results.** In Tables 2 and 3 of the initially submitted PDF, we show the experiment results of directly finetuning a small language model, which is highlighted by the grey color. The E4SRec$_{4}$ model, that has the same decoder layers, signicantly lose the performance compared to SLMRec.
>
> > ***"Q2. How do we ensure that the teacher model can transfer the recommendation-related information?"***
>
> We aim to clarify the role of the teacher model in our approach.
>
> **LLM for Rec.** Before knowledge distillation, we first train the LLMs (teacher model) on downstream recommendation datasets. This transforms the LLM into a recommendation-specific teacher model. The model and training details can refer to Figure 1 and Section 3 of our initially submitted PDF.
> During knowledge distillation, the student model learns to mimic the feature representations of the teacher model, which now contains domain-specific recommendation knowledge.
>
> Given this background, it ensures the transfer of recommendation-specific expertise rather than general NLP knowledge.
>
> > ***"Q3. Authors should add self-supervised sequential recommendation methods."***
>
> Following the reviewer's feedback, we have added three self-supervised baselines [5-7] to our evaluation. Also, we report the average ranking of each method. Our comparative analysis encompasses 11 baseline methods, spanning traditional sequential approaches, self-supervised methods, and both generation-based (G-LLMRec) and embedding-based (E-LLMRec) recommendation architectures. Furthermore, a comprehensive introduction to these three self-supervised methods is provided in the Appendix of the updated PDF.
>
> The self-supervised sequential methods demonstrate superior performance over traditional sequential methods, highlighting the effectiveness of incorporating self-supervised learning signals for enhanced sequential recommendation capabilities. Nevertheless, these approaches are inherently constrained by their limited model capacity and parameter space, resulting in performance gaps when compared to LLM-based recommendation systems.
>
> | Dataset | Cloth | Cloth | Cloth | Cloth | Movie | Movie | Movie | Movie |Rank |
> |:-------:|:-----:|:-----:|:------:|:-----:|:-----:|:-----:|:------:|:-----:|:-----:|
> | Metrics | HR@1 | HR@5 | NDCG@5 | MRR | HR@1 | HR@5 | NDCG@5 | MRR |
> | S$^3$-Rec | 14.10 | 18.67 | 16.10 | 16.95 | 7.75 | 20.39 | 15.69 | 14.34 |7.50|
> | DuoRec | 13.06 | 18.29 | 15.79 | 15.42 | 10.07 | 20.37 | 17.96 | 16.61 |7.88|
> | MAERec | 13.29 | 18.35 | 15.68 | 16.13 | 8.89 | 20.24 | 16.03 | 15.28 |8.38|
> | SLMRec$_{4\leftarrow8}$ | **16.69** | **19.47** | **18.07** | **18.74** | **15.29** | **24.25** | **19.90** | **20.36** |**1.50**|
>
> | Dataset | Music | Music | Music | Music | Sport | Sport | Sport | Sport |Rank |
> |:-------:|:-----:|:-----:|:------:|:-----:|:-----:|:-----:|:------:|:-----:|:-----:|
> | Metrics | HR@1 | HR@5 | NDCG@5 | MRR | HR@1 | HR@5 | NDCG@5 | MRR |
> | S³-Rec | 2.48 | 7.37 | 4.94 | 4.68 | 4.14 | 8.49 | 6.89 | 7.35 |6.88|
> | DuoRec | 1.84 | 4.50 | 3.19 | 3.04 | 4.13 | 8.81 | 7.03 | 6.64 |9.13|
> | MAERec | 2.19 | 6.35 | 4.67 | 3.96 | 4.01 | 8.35 | 6.65 | 6.98 |8.63|
> | SLMRec$_{4\leftarrow8}$ | **5.72** | **9.15** | **7.48** | **8.03** | **6.62** | **9.83** | **8.25** | **8.89** |**1.25**|

---

> > ### Author Response · Authors · 2024-11-20
> > **Response to Reviewer jtBY (Part 2 of 2)**
> >
> > > ***"Q4. What is the efficiency of SLMRec compared to traditional recommendation methods?"***
> >
> > To evaluate deployment efficiency in real-world scenarios, we report inference times with traditional recommendation approaches. Our experimental results demonstrate that SLMRec achieves comparable computational efficiency to traditional methods like SASRec [8] and MAERec [7], while delivering a substantial performance improvement of 45.26% over SASRec. For detailed experimental results, please refer to Table 8 in the appendix of the updated PDF.
> >
> > *Efficiency comparison of SASRec, MAERec and our SLMRec in terms of epoch-wise inference time (hours). These comparisons were conducted on a machine with an A100 GPU. During inference, models leverage parallel processing with a batch size of 512*
> >
> >
> > | Method       | Inf time(h) | Improv. (%) |
> > |--------------|-------------|-------------|
> > | SASRec      | 0.015       | 0.00        |
> > | MAERec      | 0.061       | 11.96       |
> > | SLMRec$_{4\leftarrow8}$   | 0.052       | 45.26      |
> >
> > > ***"Q5. In Table 2, the second-best performance seems inconsistent."***
> >
> > Thank you for the helpful feedback. In the initially submitted PDF, the color highlighting in Table 2 was based on the average ranking across methods within the table, rather than performance measured by individual metrics. To improve clarity, we have added a new column in the updated PDF that explicitly presents the average ranking of each method.
> >
> > Reference:
> >
> > [1] Li, Xinhang, et al. "E4srec: An elegant effective efficient extensible solution of large language models for sequential recommendation." arXiv preprint arXiv:2312.02443 (2023).
> >
> > [2] Wang, Hanbing, et al. "Rethinking large language model architectures for sequential recommendations." arXiv preprint arXiv:2402.09543 (2024).
> >
> > [3] Zhu, Yaochen, et al. "Collaborative large language model for recommender systems." Proceedings of the ACM on Web Conference 2024. 2024.
> >
> > [4] Devlin, Jacob. "Bert: Pre-training of deep bidirectional transformers for language understanding." arXiv preprint arXiv:1810.04805 (2018).
> >
> > [5] Zhou, Kun, et al. "S3-rec: Self-supervised learning for sequential recommendation with mutual information maximization." Proceedings of the 29th ACM international conference on information & knowledge management. 2020.
> >
> > [6] Qiu, Ruihong, et al. "Contrastive learning for representation degeneration problem in sequential recommendation." Proceedings of the fifteenth ACM international conference on web search and data mining. 2022.
> >
> > [7] Ye, Yaowen, Lianghao Xia, and Chao Huang. "Graph masked autoencoder for sequential recommendation." Proceedings of the 46th International ACM SIGIR Conference on Research and Development in Information Retrieval. 2023.
> >
> > [8] Kang, W. C., & McAuley, J. (2018, November). Self-attentive sequential recommendation. In 2018 IEEE international conference on data mining (ICDM) (pp. 197-206). IEEE.

---

> ### Author Response · Authors · 2024-11-26
> **Kindly Reminder**
>
> Dear Reviewer,
>
> We have submitted our detailed rebuttal addressing all raised concerns. We would greatly appreciate it if you could review our response and let us know if any aspects require further clarification.
>
> Best regards,
>
> Authors

---

> ### Author Response · Authors · 2024-11-27
> **Kindly Reminder - 2**
>
> Thank you for your reviewing. If our response indeed addresses your concerns, we'd appreciate it if you could adjust the rating accordingly.

---

### Official Review · Reviewer_HP9b · 2024-11-04

**Soundness:** 4
**Presentation:** 4
**Contribution:** 4
**Rating:** 8
**Confidence:** 4

**Summary:**

This paper introduces SLMREC, a method that enhances small language models for sequential recommendation tasks using knowledge distillation. The authors conduct extensive experiments on large-scale industry datasets, revealing that most intermediate layers in large language models are redundant. SLMREC achieves competitive performance with only 13% of the parameters of LLMs, making it highly efficient and practical for real-world applications. The model is orthogonal to other post-training efficiency techniques, such as quantization and pruning, allowing for combined use to further improve efficiency. Comprehensive experimental results demonstrate that SLMREC outperforms existing state-of-the-art methods in various evaluation metrics.

**Strengths:**

- The paper reveals that most intermediate layers in large language models are redundant, providing valuable insights into the architecture of these models.

- The paper introduces a novel method, SLMREC, which effectively enhances small language models for sequential recommendation tasks using knowledge distillation. SLMREC achieves competitive performance using only 13% of the parameters of large language models, making it highly efficient and practical for real-world applications.

- The proposed method is orthogonal to other post-training efficiency techniques like quantization and pruning, allowing for combined use to further enhance efficiency.

**Weaknesses:**

- The paper does not describe the specific process of knowledge distillation in detail, and lacks a detailed explanation of the key steps in the distillation process (such as temperature adjustment, loss function selection, etc.)

**Questions:**

- The article mentions the use of multiple supervisory signals. What are these signals? How can we combine them in the training process?

---

> ### Author Response · Authors · 2024-11-20
> **Response to Reviewer HP9b**
>
> Thank you for your positive evaluation of our work. We appreciate your recognition of our key findings regarding model redundancy, the effectiveness of SLMRec in achieving competitive performance with significantly fewer parameters, and its compatibility with other efficiency-enhancing techniques. We address the two question points that can help to further improve the clarity.
>
> > ***"Q1. The details of the knowledge distillation process."***
>
> In this work, we adopt feature-based rather than logit-based knowledge distillation, thus eliminating the need for temperature hyperparameters. Our goal is to enable the student model to learn hidden representation encoding similar to the teacher model rather than simply mimicking predictions. To achieve this, we implement two straightforward feature distillation functions: feature similarity ($D_{cos}$) and feature norm regularization ($D_{norm}$). As shown in Equations 3 and 4, these distillation functions are applied after each block of multiple layers, enabling fewer layers to extract feature representations as effectively as many layers.
>
> > ***"Q2. What are multiple supervisory signals? How do we combine them in the training process?"***
>
> Multiple supervisory signals are designed to guide the student model in learning specific aspects of recommendation-related knowledge. For each block, we introduce prediction layers that compute user-item prediction scores from the feature representations, optimized using cross-entropy loss (Eq. 1). During training, these multiple supervisory signals work in conjunction with the two feature distillation loss functions to optimize the student model, as formulated in Eq. 6. For clarity, we have revised Figure 3 of the updated PDF to better illustrate the multiple supervisory signals.

---

> ### Author Response · Authors · 2024-11-26
> **Kindly Reminder**
>
> Dear Reviewer,
>
> We have submitted our detailed rebuttal addressing all raised concerns. We would greatly appreciate it if you could review our response and let us know if any aspects require further clarification.
>
> Best regards,
>
> Authors

---

> > ### Comment · Reviewer_HP9b · 2024-11-26
> > **Thanks for the response**
> >
> > Thanks for the authors' response. My concerns have been addressed.

---

> > > ### Author Response · Authors · 2024-11-26
> > >
> > > We greatly appreciate your review and are pleased that we could address all your concerns satisfactorily. Thank you for your valuable feedback.

---

### Author Response · Authors · 2024-11-24
**General Response by Authors**

Dear Area Chairs and Reviewers,

We appreciate the reviewers' time, valuable comments, and constructive suggestions.

---

**Strengths** acknowledged by the reviewers:

1. Key Contributions: identifying redundant layers in large language models (Reviewer **HP9b**) and developing SLMRec with competitive performance using only 13% parameters (Reviewers **HP9b**, **M71Q**).
2. Well-written (Reviewers **jtBY**, **ZHQw**); Strong theoretical justification (Reviewer **jtBY**); Clear illustrations (Reviewer **jtBY**).
3. Comprehensive experiments (Reviewer **M71Q**) with strong empirical validation; Practical value for real-world deployment (Reviewers **M71Q**, **ZHQw**).
4. Strong theoretical justification demonstrating why small language models can achieve comparable user representation learning capabilities (Reviewer **jtBY**).

---

Reviewers raise three main concerns:

1. Details about the knowledge distillation (Reviewer **HP9b**, **M71Q**).
2. More comparison baselines (Reviewer **jtBY**).
3. Misunderstanding of the motivation (Reviewer **jtBY**) and the novelty (Reviewer **ZHQw**).

These primary concerns have been successfully addressed during the rebuttal phase, and we hope that the improvements we made during this stage will be considered.

---

Revisions in the rebuttal phrase:

1. Added Experimental Comparisons: We have incorporated three additional self-supervised sequential recommendation baselines to strengthen our comparative analysis.
2. Enhanced Results Presentation: We have highlighted the average ranking of each method to provide clearer performance comparisons.

All these revisions have been highlighted in blue in the updated PDF.

---

We sincerely appreciate your valuable time!

Thanks and regards,

Authors

---

### Meta-Review · Area_Chair_vpLY · 2024-12-22

**Metareview:**

This paper focuses on empowering small language models for sequential recommendation and proposes the SLMRec model for sequential recommendation tasks, mainly using knowledge distillation strategies to transfer recommendation capabilities to smaller LLM models with fewer layers. Most reviewers recognized the value of this research, but some weaknesses were raised, including: 1) The details and motivation behind the distillation process were not fully explained, and all reviewers mentioned this deficiency; 2) The original paper lacked baseline comparisons. During the rebuttal phase, the authors addressed these issues and resolved most of them. After reviewing the feedback and the original paper, I think that overall, this work is valuable but needs further improvements based on the reviewers' suggestions. Additionally, I personally found it unclear why the model's performance with 31 layers in Figure 2 is particularly poor.

**Additional Comments On Reviewer Discussion:**

The authors replied to the reviewers, but most of them did not provide more feedback/questions.

---

### Decision · Program_Chairs · 2025-01-22

Accept (Poster)